# Extending nnU-Net for the KiTS21 Challenge

Alex Markow[1], Lea Sophie Heinzelmann[1], Luca Emanuel Fioravanti[1], Matthias Ngo[1] and Richard Hempel[1]

[1]Hochschule Furtwangen University (HFU)
alex.markow@hs-furtwangen.de
lea.sophie.heinzelmann@hs-furtwangen.de
luca.emanuel.fioravanti@hs-furtwangen.de
matthias.ngo@hs-furtwangen.de
richard.hempel@hs-furtwangen.de

**Abstract.** Tools for automatic semantic segmentation of 3D CT scans have shown significant results for kidney and tumour extraction using deep learning methods. The ability to produce reliable and accurate segmentations will help with surgical planning and improve the efficiency of malignant tumour detection. In this paper, we describe a three-stage pipeline for kidney, tumour and cyst segmentation based on the nnU-Net framework [1]. The initial stages extract the kidney regions and detect class presents. Then the last stage individually segments the detected classes in the cropped areas.

**Keywords:** KiTS21 Challenge, U-Net, Medical Image Segmentation

*As of the submission on the 23rd of August, our paper is incomplete. A complete submission will be provided shortly, in which all sections will be expanded upon. This version was submitted only to state our intention to take part in the KiTS21 challenge.*

## 1 Introduction

During malignant tumour treatment in the kidney regions, CT scans are analysed for pre-surgical planning to determine the extent of a tumour. But the evaluation of renal cancer can be difficult and time-consuming, because of kidney and kidney tumour morphology [2, 3]. This is one of the major reasons for development of automated tooling, which can create accurate segmentation of these medical images. The KiTS21 challenge attempts to speed up the development of such tools. The challenge provides participants with a dataset of 300 scans and annotations. Performance is ranked using the Sørensen-Dice and Surface Dice [4] scores. As an extension of the previous KiTS19 challenge, KiTS21 includes the Surface Dice metric and the cyst class for segmentation additionally to the kidney and tumour classes. Given the prevalence of U-Net variants in the high-ranking positions last challenge [5], our approach focuses on the development of such a variant based on the nnU-Net [1] adapted for this year's challenge.

## 2 Methods

The segmentation classes make up only a small section of the images and therefore present a particular difficulty during detection. This problem has been especially observed for the cyst class. To counteract this, we designed a three-stage method using the nnU-Net framework [1]. The first stage crops the image down to its kidney regions. The second stage detects class presents to reduce false positives and unnecessary computations in the third stage. Then the segmentation for the present classes in the cropped areas is performed during the last stage.

### 2.1 Training and Validation Data

Our submission made use of the official KiTS21 training set alone.

### 2.2 Preprocessing

Our preprocessing scheme follows the default preprocessing path provided by the nnU-Net framework.

## 3 Results

The hardware, used for the training of the networks, consisted of an AMD Ryzen Threadripper 1920x CPU with 106 GB RAM and a NVIDIA GeForce RTX 2080 TI with 11 GB VRAM.

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
