# OpenReview forum: "Extending nnU-Net for the KiTS21 Challenge"
_MICCAI.org/2021/Challenge/KiTS — Submitted to KiTS21 Challenge_

### Official Review · Reviewer_6nCe · 2021-08-30

**Rating:** 4

**Review:**

Broadly, the authors appear to be describing a coarse-to-fine approach to the challenge. This is a largely unfinished paper submission and serves only to declare this teams intention to submit.

---

### Official Review · Reviewer_cCUs · 2021-08-30

**Rating:** 3

**Review:**

### Overall

- It's fine that the paper is incomplete for this initial submission. Please do expand it and resubmit as soon as possible so that we can re-review it

### Introduction

- The first statement applies to all kidney tumors

### Methods

- It would be nice to include a diagram that summarizes your approach. This always serves to improve clarity
- Section is unfinished

### Results

- It would be nice to include a figure that shows an example of your method's predictions vs the ground truth for that case
- Section is unfinished

### Discussion and Conclusion

- Section is unfinished/not included. Please make sure to include this section in your revision

---

### Decision · Program_Chairs · 2021-08-30

**Decision:**

Major Revisions

**Comment:**

Please address the reviewer comments and resubmit